# Finding New Molecular Targets of Familiar Natural Products Using In Silico Target Prediction

**DOI:** 10.3390/ijms21197102

**Published:** 2020-09-26

**Authors:** Fabian Mayr, Gabriele Möller, Ulrike Garscha, Jana Fischer, Patricia Rodríguez Castaño, Silvia G. Inderbinen, Veronika Temml, Birgit Waltenberger, Stefan Schwaiger, Rolf W. Hartmann, Christian Gege, Stefan Martens, Alex Odermatt, Amit V. Pandey, Oliver Werz, Jerzy Adamski, Hermann Stuppner, Daniela Schuster

**Affiliations:** 1Institute of Pharmacy/Pharmacognosy, Center for Molecular Biosciences Innsbruck (CMBI), University of Innsbruck, Innrain 80/82, 6020 Innsbruck, Austria; F.Mayr@uibk.ac.at (F.M.); Veronika.Temml@pmu.ac.at (V.T.); birgit.waltenberger@uibk.ac.at (B.W.); Stefan.Schwaiger@uibk.ac.at (S.S.); hermann.stuppner@uibk.ac.at (H.S.); 2Research Unit Molecular Endocrinology and Metabolism, Helmholtz Zentrum München, Ingolstädter Landstraße 1, 85764 Neuherberg, Germany; gabriele.moeller@helmholtz-muenchen.de (G.M.); adamski@helmholtz-muenchen.de (J.A.); 3Department of Pharmaceutical/Medicinal Chemistry, Institute of Pharmacy, University Greifswald, Friedrich-Ludwig-Jahn-Straße 17, 17489 Greifswald, Germany; ulrike.garscha@uni-greifswald.de (U.G.); jana.fischer@uni-greifswald.de (J.F.); 4Pediatric Endocrinology, Diabetology and Metabolism, University Children’s Hospital Bern, Freiburgstrasse 15, 3010 Bern, Switzerland; patrirodcas@gmail.com (P.R.C.); amit.pandey@dbmr.unibe.ch (A.V.P.); 5Department of Biomedical Research, University of Bern, Freiburgstrasse 15, 3010 Bern, Switzerland; 6Division of Molecular and Systems Toxicology, Department of Pharmaceutical Sciences, University of Basel, Klingelbergstrasse 50, 4056 Basel, Switzerland; silvia.inderbinen@unibas.ch (S.G.I.); alex.odermatt@unibas.ch (A.O.); 7Helmholtz Institute of Pharmaceutical Research Saarland (HIPS), Department for Drug Design and Optimization, Campus E8.1, 66123 Saarbrücken, Germany; rolf.hartmann@helmholtz-hzi.de; 8Saarland University, Pharmaceutical and Medicinal Chemistry, Campus E8.1, 66123 Saarbrücken, Germany; 9University of Heidelberg, Institute of Pharmacy and Molecular Biotechnology (IPMB), Medicinal Chemistry, Im Neuenheimer Feld 364, 69120 Heidelberg, Germany; christian.gege@web.de; 10Research and Innovation Centre, Fondazione Edmund Mach (FEM), Via Mach 1, 38010 San Michele all’Adige, Italy; stefan.martens@fmach.it; 11Department of Pharmaceutical/Medicinal Chemistry, Institute of Pharmacy, Friedrich-Schiller-University Jena, Philosophenweg 14, 07743 Jena, Germany; oliver.werz@uni-jena.de; 12Lehrstuhl für Experimentelle Genetik, Technische Universität München, Emil-Erlenmeyer-Forum 5, 85356 Freising-Weihenstephan, Germany; 13Department of Biochemistry, Yong Loo Lin School of Medicine, National University of Singapore, 8 Medical Drive, Singapore 117597, Singapore; 14Institute of Pharmacy, Department of Pharmaceutical and Medicinal Chemistry, Paracelsus Medical University Salzburg, Strubergasse 21, 5020 Salzburg, Austria; 15Institute of Pharmacy/Pharmaceutical Chemistry, Center for Molecular Biosciences Innsbruck (CMBI), University of Innsbruck, Innrain 80/82, 6020 Innsbruck, Austria

**Keywords:** in silico target prediction, dihydrochalcones, SEA, SwissTargetPrediction, SuperPred, polypharmacology, virtual screening

## Abstract

Natural products comprise a rich reservoir for innovative drug leads and are a constant source of bioactive compounds. To find pharmacological targets for new or already known natural products using modern computer-aided methods is a current endeavor in drug discovery. Nature’s treasures, however, could be used more effectively. Yet, reliable pipelines for the large-scale target prediction of natural products are still rare. We developed an in silico workflow consisting of four independent, stand-alone target prediction tools and evaluated its performance on dihydrochalcones (DHCs)—a well-known class of natural products. Thereby, we revealed four previously unreported protein targets for DHCs, namely 5-lipoxygenase, cyclooxygenase-1, 17β-hydroxysteroid dehydrogenase 3, and aldo-keto reductase 1C3. Moreover, we provide a thorough strategy on how to perform computational target predictions and guidance on using the respective tools.

## 1. Introduction

Finding new chemical entities that alter a biological response—the quintessence of drug discovery—is a constant endeavor in pharmaceutical science. In contrast, the need for novel, improved clinical candidates has also remained consistently high, urging drug discovery scientists to explore fresh ground. The integration of chemoinformatic and bioinformatic tools into drug discovery in the early 1990s and the recent advances in big data handling have leveraged access to a myriad of massive public datasets [1,2] and powerful tools, e.g., virtual screening (VS) [3]. In the past decade, the concept of drug repurposing has emerged as an attractive strategy to rededicate approved drugs or partially developed compounds to new molecular targets [4,5]. This development is, next to the intention to reduce R&D costs, also owed to advances in computational chemistry [6]. The latter can be achieved by a so-called “inverse VS” utilizing techniques like 2D-similarity searches [7], 3D-similarity searches [8], and pharmacophore-based VS [9,10]. Many such tools have been made public in the past decade [11,12,13], aiming to boost both drug repurposing efforts and drug discovery as a whole.

Natural products are remarkable in many regards, particularly for being the main source of drugs in the past and, today, by serving as a source for innovative leads [14]. Natural products bear privileged structural features that were “shaped” by evolution, yielding compounds that can serve as promising starting points for drug development [15,16]. Further, natural products often show polypharmacological properties, interacting with more than one target [17]. The two groups around Gisbert Schneider and Stuart L. Schreiber found that natural products are more likely to act as true polypharmacological agents rather than unspecific binders—a property instead associated with synthetic compounds [18,19].

Based on this concept, tools or workflows that allow for the accurate prediction of new molecular targets for natural products are of great interest. We here propose a workflow to specifically search for yet unreported protein targets of known compounds, using a combination of in silico and in vitro methods (Figure 1). The compounds of interest are virtually screened against our in-house resources, representing a panel of 39 drug targets expressed as 387 pharmacophore models (a comprehensive list of all models is provided in Appendix A). Additionally, the compounds are subjected to three independent open-access target prediction servers. The results generated by these four diverse methods are combined, and those with the highest consensus (protein targets predicted by several methods independently of each other) are selected for in vitro evaluation. Moreover, already known protein targets are excluded from further investigations by checking their appearance in PubChem (https://pubchem.ncbi.nlm.nih.gov/), a pertinent open knowledge base for bioactivities [20].

We evaluated our workflow for fitness by performing a target prediction of a complete class of established natural products, namely dihydrochalcones (DHCs). DHCs are readily available from nature, since several representatives are highly accumulated in both the fresh and withered leaves of apple trees [21]. On the other hand, phloridzin, one of the most frequently found DHCs, served as the lead structure for the development of sodium/glucose cotransporter 2 (SGLT2) inhibitors like dapagliflozin, approved drugs for the treatment of type 2 diabetes [22]. DHCs have recently returned into focus, since their descendants, in clinical use now for about eight years, have shown therapeutic benefits that go beyond SGLT2 inhibition, like, e.g., in heart failure [23]. This instance points towards a high polypharmacological potential of the drugs and its parental template. However, phloridzin research has so far been focused on its antidiabetic, antioxidative, and estrogenic effects. Thus, we prepared a comprehensive virtual library of DHCs (naturally occurring ones and those with modest semisynthetic modifications), predicted and selected promising, potentially new DHC targets, and tested ten common DHCs in respective in vitro assays.

## 2. Results

### 2.1. Data Basis, Curation, and Technical Setup of In Silico Predictions

To realistically mirror the true diversity of DHCs, we gathered 425 DHCs from the literature that were either naturally occurring or roughly resembled physicochemical properties of natural DHCs (molecular weight, ratio glycosides/aglyca, and physicochemical properties). Accordingly, we called this virtual library the “DHC chemical space” (see Figure 1, Data Preparation). A panel of ten commonly found DHCs (**1**–**10**, see Table 1) that were physically available to us and intended for in vitro testing were also included in the DHC chemical space. The DHC chemical space was then screened against our historically grown pharmacophore model database (Ph-DB), and the results were written to a matrix (e.g., compound **1** is predicted to act on protein A). The DHC chemical space was, in parallel, also subjected to the three target prediction servers: Similarity Ensemble Approach (SEA), SwissTargetPrediction (STP), and SuperPred (SP), each of them predicting potential targets for each of the 425 DHCs. All results were combined into one matrix called the “predicted DHC biological space” (see Figure 1, Virtual Screening). In addition, all of the known bioactivities of the 425 compounds in the DHC chemical space were downloaded from PubChem and the resulting matrix called the “known DHC biological space” (see Figure 1, Bioactivity Mining). The known DHC biological space was additionally depicted as a network, as shown in Figure 2. Next, the compound-target interactions present in the known DHC biological space were removed from the predicted DHC biological space, and a CS was assigned to each prediction (see Figure 1, Scoring and Selecting). The CS is an expression on how high the consensus of a prediction is, in addition to a positive prediction by our Ph-DB. We introduced the requirement of a hit with our own models, since they are well-validated, and for most targets, experimental testing of the hits was available. For instance, if compound 1 was predicted to bind to protein A by our in-house pharmacophore models and two further target prediction servers, the CS was three. Finally, based on the CS and other criteria, the six most promising protein targets were selected, and compounds **1**–**10** were assayed in vitro. 

To automate the in silico part of the workflow, operations made for screening our Ph-DB, submission and reconciliation to target prediction servers, and bioactivity mining were performed via custom-made scripts. All input and output files generated in this workflow, including the scripts and a corresponding Jupyter Notebook containing all data manipulations, were provided via GitHub (https://github.com/fmayr/DHC_TargetPrediction). For better clarity, the relationships and data flow are schematically shown in Appendix A.

### 2.2. Predicted and Unknown DHC Biological Space

The final selection of targets to be evaluated in vitro was made based on four criteria (summarized for twelve frequently predicted targets in Table 2). First, the interactions of **1**–**10** that were predicted by Ph-DB and had a CS of two or higher (see Figure 3) were included (Table 2, Selection Criterion I). Thereby, 5-lipoxygenase (5-LO) was highlighted for **4** (CS = 2) and **5** (CS = 3), and aromatase was highlighted for **1**, **5**, **6**, **7**, **8**, and **9** (CS = 2). Figure 3 also shows that a handful of other targets achieved high CSs, e.g., acetylcholinesterase (AChE), estrogen receptor α (ERα), protein-tyrosine phosphatase B1 (PTP1B), and nuclear factor κB (NF-κB). However, ERα and NF-κB were neglected, since they were already reported targets for at least one DHC [24,25], while suitable assays for AChE and PTP1B were not available. 

Second, protein targets that were particularly frequently predicted for the whole DHC chemical space (425 compounds) with a positive prediction of Ph-DB and CSs of two or higher (Table 2, Selection Criterion II) were included. We hypothesized that these targets were generally well-suited for the DHC scaffold. 17β HSD2 and 17β HSD3 were the fourth and fifth most frequently predicted targets with a CS of three, behind AChE (assays not available), ERα (already established target for some DHCs), and 5-LO (already selected for in vitro testing; see Appendix A). Moreover, cyclooxygenase 1 (COX-1) and aldo-keto reductase 1C3 (AKR1C3) were the fourth and sixth most frequently predicted targets, respectively, with CSs of two (see Appendix A). Higher ranked targets were all either already known (ERα and ERβ) or already included in our selection (aromatase and 5-LO). 

Third, the overall predictions were evaluated for their novelty, their consistency, and whether our approach could produce high scores for known ligand-target interactions (Table 2, Selection Criterion III). Novelty means that only unreported targets were selected, while prediction consistency means that predicted targets are more credible if they are biologically related to one another, e.g., isoenzymes or proteins that belong to the same pathway. It is actually oftentimes the case that one compound binds to several closely related targets (lack of specificity), which should be reflected in the virtual predictions [26,27]. Further, great value is added if, e.g., a closely related target was already reported or if known targets are enriched in the predictions generated by the in silico workflow. In our case, we observed targets belonging to the steroid metabolism (aromatase, 17β HSD2, 17β HSD3, and AKR1C3) or to arachidonic acid (AA) metabolism (5-LO and COX-1). Targets of the steroid metabolism are obviously closely related to ERs; aromatase, 17β HSD2, and ERs even share the same substrate or ligand, respectively, namely estradiol. The shape and pharmacophore of those targets’ binding sites must, therefore, be somewhat similar. On this basis, we hypothesized that the four predicted targets of the steroid metabolism are promising DHC targets, given that ERs are confirmed targets of several DHCs. Similarly, COX-1 and 5-LO share the same substrate, namely AA, implying that, also, the binding sites of the latter two must be similar to a certain extent. Moreover, **7** was confirmed to inhibit 15-hydroxyprostaglandin dehydrogenase and soluble epoxide hydrolase—two enzymes in the AA pathway related to COX-1 and 5-LO with, again, presumably similar binding sites (see Figure 2). Steroid metabolism and AA metabolism are on their side interconnected by AKR1C3, which is also commonly referred to as 17β-hydroxysteroid dehydrogenase 5 or prostaglandin F synthase (see Appendix A). Indeed, this enzyme converts both steroids and AA-like fatty acids using the same binding site [28]. From there, it was concluded that those six protein binding sites may share substantial similarities, and the predictions of the latter can be considered consistent. Finally, we checked if targets of the known DHC biological space could be enriched by our target prediction workflow. Effectively, we observed a clear enrichment of the consensus scored target frequencies (see Appendix A) compared to the stand-alone target prediction tools (see Appendix A).

Fourth, the availability of a suitable assay was logically a pivotal criterion for targets to be selected (Table 2, Selection Criterion IV).

### 2.3. New DHC Biological Space

Following the definition of the six protein targets issued for biological evaluation, the respective in vitro assays were performed using compounds **1**–**10** (see Figure 4 and Table 3). Biological activities were expressed as the percent inhibition at a 10-µM compound concentration relative to the mock control (= 0%). Three independent experiments (*n* = 3) were conducted, and the mean inhibition plus/minus standard deviation is depicted. Mean inhibition values that were below 30% were regarded as inactive; negative inhibition values and relative standard deviations larger than 20% were regarded as ambiguous assay results and, thus, as inactive. 17β HSD2 measurements yielded ambiguous assay results for all compounds, which were regarded as inactive, as well as aromatase, where all ten compounds were inactive. For 5-LO, COX-1, AKR1C3, and 17β HSD3, at least one of the ten compounds exhibited weak inhibitory activities towards the respective target. Thus, **4** and **5** showed weak inhibition of AKR1C3; **2** showed weak inhibition of 17β HSD3; and **1**, **2**, **3**, and **5** showed weak inhibition of COX-1. The ten DHCs showed the best results of this study in 5-LO inhibition, where **1**, **2**, **7**, **8**, and **10** showed weak-to-moderate inhibitory activities, **1** and **2** even reached mean inhibition values of 85.4% ± 9.3% and 99.2% ± 1.2%, respectively. In the course of this study, six further protein targets were evaluated in vitro due to availability of the respective assays, rather than based on in silico predictions. Most of the measured activities were not affected, however, to not withhold those results to the community; the results are shown in Appendix A.

## 3. Discussion

In the current study, we present an in silico target prediction workflow capable of prioritizing new molecular targets for known chemical entities, here exemplified on DHCs. Even though in silico target prediction is common today for synthetic compounds [7,11,30,36], it is still challenging for natural products [37,38,39,40]. To our knowledge, this is the first study performing in silico target prediction on a natural product class while systematically combining diverse established tools. We recently observed that, especially when predicting targets for natural products, none of the established tools (including our own) performed perfectly, but owing to different methods, all of them performed differently [41]. Exploiting the complementarity of the methods’ strengths should, therefore, correct for this shortcoming and result in a better predictive performance. This strategy might be of great interest to the community, since it underlines the benefits of predicting targets of a closely related compound series rather than single compounds, and it provides a thorough use case of publicly available tools and how to interpret its predictions. Additionally, we recently predicted DHCs to be inhibitors of the mushroom tyrosinase, albeit they turned out to be alternative substrates of the latter—an unexpected form of bioactivity that an in silico target prediction cannot distinguish from competitive inhibition [42].

Finally, six potential DHC targets were selected—of which, four could be experimentally confirmed (5-LO, COX-1, 17β HSD3, and AKR1C3) as molecular targets of at least one of the ten DHCs **1**–**10**. These four proteins are all new DHC targets and, thus, expanded the DHC biological space, which we were aiming for (see Figure 2). In terms of accuracy, our yield of novel targets (four out of six) is clearly superior to the yield of a “random selection” (test a random panel of protein targets towards compounds **1**–**10**), which typically lies below 1% [43,44,45,46]. 17β HSD2 and aromatase could not be confirmed as targets; however, a thorough literature search revealed that **1** was reported once in the literature to inhibit aromatase with an IC_50_ value ≥ 50 µM [47]. Activities at such high concentrations are considered inactive by PubChem and were thus not retrieved by our bioactivity mining approach. This activity is indeed negligible; however, this value seems consistent with the results generated by us (13.8 ± 2.0% inhibition at 10 µM, see Table 3). The observed activities were all in typical ranges for the nonoptimized lead structures discovered by VS. Ripphausen et al. conducted a survey in 2010 showing that hit compounds identified by VS have defined potency endpoints (IC_50_, EC_50_, K_i_, or K_d_) of 4 to 19 µM in average, which is arguably high. However, the true value of VS lies in the ability to identify new chemotypes as leads or, in its turn, to identify new targets for known compounds in target prediction [48]. The exciting fact of this study is that our workflow could successfully prioritize new molecular targets of well-known compounds with noteworthy accuracy, opening up new avenues for DHC research (see Appendix A).

The results can now be utilized in manifold ways. First, our findings revealed polypharmacology of some DHCs as a sideline. Thus, **1** and **2** seem to have promising anti-inflammatory properties by simultaneously inhibiting 5-LO and COX-1. Investigating whether this property is inherent to gliflozin drugs could be the subject of another study. Potential anti-inflammatory properties of gliflozin drugs could contribute to the currently observed beneficial effects of those drugs in, e.g., heart failure [49]. Several authors have described the anti-inflammatory properties of gliflozin drugs on a functional level; however, a distinct mode of action and association to distinct molecular targets is still to be elucidated [50,51,52].

Second, to remain with polypharmacology, e.g., **4** and **5** could serve as leads to develop multitarget inhibitors directed against AKR1C3 and 5-LO. Analogously, **2** could be used as a starting point for the development of a multitarget inhibitor directed against 17β HSD3 and 5-LO. Following lead and selectivity optimization, both options would represent a new compound class with potentially interesting pharmacological properties. These agents could be beneficial, e.g., in the treatment of prostate cancer, since AKR1C3 and 17β HSD3 are frequently linked to this condition [53,54], and malignant tumors are usually surrounded by a proinflammatory microenvironment [55]. Pursuing this very goal, we investigated other natural benzylated DHCs in a related study that was recently published. Those compounds came to our attention for their good AKR1C3 and microsomal prostaglandin E synthase (mPGEs-1) inhibiting properties—another enzyme involved in the synthesis of proinflammatory AA products. The compound showed prominent inhibitory activity on prostate cancer cell proliferation—interestingly, even in sophisticated 3D coculture models, which mimic the physiological state of the tumor more accurately. After elucidating the mode of action, it was found that our DHC acts by inhibiting AKR1C3 and by antagonizing the androgen receptor noncompetitively, thus exerting strong antiandrogenic signaling in the prostate cancer cell. Apart from the just-mentioned investigation on prostate cancer, several other similar scenarios for utilization of the presented results are thinkable. 

Third, our target prediction workflow revealed the great potential of DHCs as lead structures. As mentioned above, DHCs are incredibly well-accessible from biomass like apple leaves in concentrations up to 20% of dry mass [56]. According to the principle of bioprospection, meaning the harnessing of resources from nature for medical purposes, apple leaves could play an interesting role in future lead optimization campaigns.

Fourth, the established in silico prediction workflow can be applied to several other compound classes than DHCs, even to much more complex structures. The limit of the applicability is the similarity of the query compounds to the reference compounds used by the prediction platforms (known actives from the literature). Compounds that significantly differ from the currently known biologically active chemical space, e.g., compounds with a very high molecular weight, may not be identified as virtual hits by the workflow. 

In summary, this study blueprints a strategy to predict targets for known compounds by mostly using open-source platforms, thereby empowering a great number of researchers to actively rededicate their compounds. This technology, however, is greatly enabled by the growing body of known bioactivities, hinting towards increasing accuracies reachable with target prediction campaigns in the near future, as this knowledge keeps on expanding.

## 4. Materials and Methods

### 4.1. Dataset Assembly

A dataset representing the chemical space of DHCs was gathered, containing naturally occurring DHCs, as well as semisynthetic derivatives of the latter. First, compounds **1**–**10** were included. Then, a thorough literature search was conducted in SciFinder (https://ucsd.libguides.com/scifinder, La Jolla, CA, USA), using a substructure search function for 1,2-diphenyl-propan-1-one. The SciFinder query was conducted on the 8th of August 2017 and yielded 5457 DHCs. The resulting compounds were checked manually for their origin (natural product vs. semisynthetic), created in ChemDraw Professional (version 16.0.0.82 (68), PerkinElmer, Waltham, MA, USA) with assigned stereochemistry and double-checked with SciFinder using the “search SciFinder” function in ChemDraw. For compounds, especially natural products, with lacking absolute configurations, all possible stereoisomers were included. Finally, the dataset “DHC_full” contained 425 natural or semisynthetic DHCs (corrected for stereochemistry) and was converted with a custom-built Pipeline Pilot protocol (version 9.5.0.831, Dassault Systèmes BIOVIA, Vélizy-Villacoublay, France) to an sd-file (DHC_full.sdf) and a csv-file (DHC_full.csv). The Pipeline Pilot protocol is shown in Appendix A. The files are provided on GitHub (see Appendix A).

### 4.2. Bioactivity Mining

The mining of bioactivity data of DHCs was done through a script coded in Python 3 (version 3.7.3, https://www.python.org/, accessed on 30 September 2019) called “bioactivity_network_generator_SMILES.py”. This script iterates over a two-column csv table (named SMILES) while fetching one SMILES code at a time. The script interacts with various application programming interfaces (APIs) of PubChem (PUG REST, https://pubchem.ncbi.nlm.nih.gov/; accessed on 30 September 2019) [20], UniProt (https://www.uniprot.org/, accessed on 30 September 2019) [57], and Reactome (https://reactome.org/, accessed on 30 September 2019) [58]. At the first stage, the SMILES code is posted as a query to the PubChem API, which returns the desired PubChem compound ID (CID), if present. The CID is then used to post a second query to the PUG REST API, fetching all assay IDs (AIDs) associated with the posted CID, for which the assay result was flagged as “active”. The produced list of AIDs is then annotated with the respective gene names from PubChem. This step eliminates all AIDs that are not associated to a single protein (e.g., cell-based assays). The gene name from PubChem can be translated into UniProt names and the respective entry names used in UniProt using the UniProt KB API. In the last step, the UniProt entry name was used to retrieve the associated human pathways from the Reactome API. The gathered bioactivity data of all provided SMILES codes was processed in Pandas (version 0.24.2, https://pandas.pydata.org/) [59] and, finally, written to a csv file in the simple interaction file (sif) format, which can be visualized in, e.g., Cytoscape (https://cytoscape.org/, accessed on 30 September 2019) [60]. Interaction weights were calculated by using the “group by” method implemented in Pandas chained with the “count()” method. Interaction weights, thus, are an integer representing the appearances of a particular interaction in PubChem. The script “bioactivity_network_generator_SMILES.py” is provided on GitHub (see Appendix A).

### 4.3. Pharmacophore-based Parallel Virtual Screening

Pharmacophore-based parallel VS was performed using the historically grown in-house pharmacophore model databased built and maintained by Prof. Daniela Schuster. The database consists of 387 ligand-based and structure-based pharmacophore models for 39 protein targets built in two different software environments, namely Discovery Studio (version 4.5.0.15071, Dassault Systèmes BIOVIA, Vélizy-Villacoublay, France) and various versions of LigandScout (Inte:Ligand, Vienna, Austria). For the Discovery Studio models, parallel screening was performed using the “ligand profiler” protocol (for settings, refer to Appendix A). Screening of the LigandScout models was performed using the “iscreen.exe” program via a command line and the databases created for every LigandScout version by using “idbgen.exe” via a command line for the respective LigandScout versions. Omega (OpenEye Scientific Software, Santa Fe, NM, USA) with the “best” settings was used. The targets represented by these models are predominantly targets belonging to the arachidonic acid (AA) cascade, as well as corresponding to downstream signaling and steroid metabolism and signaling. The targets are often associated with inflammation, neoplasm, or are popular off targets. A detailed compilation of all models is provided in Appendix A.

### 4.4. Target Prediction with Publicly Available Tools

Next to the pharmacophore-based parallel screening, three target prediction tools that are available as web servers were used, namely SEA (http://sea16.docking.org/; accessed on 18 September 2019) [30], STP (http://www.swisstargetprediction.ch/; 18.09.2019) [32,61,62], and SP (http://prediction.charite.de/; accessed on 18 September 2019) [31,63]. SEA is a 2D ligand-based similarity ensemble method. Each target present in SEA is described by a set of its known ligands of various sizes. An input ligand is then compared against all ligands of all target sets via Tanimoto similarity of the ECFP4 fingerprints. For each target, the Tanimoto similarities are summed up and z-scores calculated. Since the authors computed the distribution of z-scores obtained between random similarity ensembles, the z-scores of a screening ligand to each target can be used to calculate the expectation values (E-values). Those E-values, similar as in the BLAST algorithm, express the likelihood that the observed similarity happened due to coincidence. SEA uses bioactivity data derived from ChEMBL [64] and is maintained by the University of California, San Francisco (UCSF). SP operates in a very similar way to SEA, being a 2D similarity ensemble approach and using ECFP4 fingerprints as well. Bioactivities that were used to build reference target sets in SP were derived from ChEMBL [65], Binding DB [66], and SuperTarget [67,68]. SP is maintained by the structural bioinformatics group of the Charité—University Medicine Berlin in Germany. In contrast to SEA and SP, STP makes use of the ligand-based similarity ensembles principle as well; however, it is a hybrid method between 2D and 3D. Two-dimensional similarity is computed via Tanimoto similarity using FP2 fingerprints, while 3D similarity is described as the Manhattan distance between the electroshape [69] vectors. Finally, a logistic regression classifies the input ligand based on 2D and 3D similarities. STP derived its bioactivity data also from ChEMBL. It was developed by the Swiss Institute of Bioinformatics (SIB). The online servers described above were accessed via a web scraper script called “TarPredCrawler.py” written in python 3. The script uses selenium (https://www.seleniumhq.org/, version 3.141.0) to send post requests of SMILES codes to the four servers and downloads the resulting predictions as a table. The script “TarPredCrawler.py” is provided on GitHub (https://github.com/fmayr/DHC_TargetPrediction).

### 4.5. Biochemical Assays

Aromatase assays were performed as previously described [70]. Briefly, genes for human wild-type aromatase and NADPH P450 oxidoreductase were transfected into *Escherichia coli* to express both proteins in the recombinant form, and proteins were purified using multiple chromatographic procedures, as described previously. Liposomes containing both enzymes were formed for the assay of enzymatic activities. Aromatase activity was quantified by measuring the release of tritiated water after incubation with 1β-^3^H androstenedione, a method introduced by Lephart and Simpson [71]. Ten-nanomolar anastrozole (CAS: 120511-73-1) were used as the positive control. 

Inhibitory activities towards AKR1C3, 17β HSD3, and 17β HSD2 were assayed as described in Schuster et al. [35]. Briefly, AKR1C3 and 17β HSD2 were transformed into *E. coli* BL21 (DE3), and 17β HSD3 was transfected into HEK293 cells. For assaying inhibitory activities towards 17β HSD2, bacterial suspensions were used, while cell suspensions were used to assay 17β HSD3, and bacterial lysates were used to assay AKR1C3. The enzyme-containing lysates or suspensions were incubated with tritiated substrates and cofactors (21-nM 17β-estradiol (6,7-^3^H) and 750-nM NAD^+^ for 17β HSD2or 10.6-nM 4-androstene-3,17-dione (1,2,6,7-^3^H), and 600-µM NADPH for 17β HSD3 and AKR1C3) in the presence of test compounds in a final concentration of 10 µM (compounds supplied in DMSO; 1% final DMSO in the assay). After a defined incubation time, substrates and products were extracted using solid-phase extraction (SPE) and analyzed by reversed phase high performance liquid chromatography (RP-HPLC) and online scintillation counting. Quantification of relative conversion occurred via chromatographic peak integration and the percentage of inhibition was calculated relative to a mock control (1% DMSO). As positive controls, compound **2–9** (CAS: 745028-76-6) [35] was used for AKR1C3 assays, compound **24** (CAS: 873206-61-2) [34] for 17β HSD3 assays, and compound **19** (CAS: 1340482-23-6) [72] for 17β HSD2 assays, all in 1-µM concentrations. Inhibitory activities towards 5-LO and COX-1 were determined as described earlier by Schaible et al. [73]. and Koeberle et al. [74], respectively. Briefly, polymorphonuclear leukocytes (for 5-LO) and human platelets (for COX-1) were freshly isolated from the blood of healthy volunteers, preincubated with the potential inhibitors, and stimulated with 2.5-μM Ca^2+^-ionophore A23187 or arachidonic acid, respectively. The reaction was stopped and substrates and products isolated and analyzed on RP-HPLC. The 5-LO products included LTB_4_, its trans-isomers, 5-HPETE, and 5-HETE, while the COX-1 product was quantified as 12-HHT. Again, quantification occurred via chromatographic peak integration and the percentage calculated relative to a mock control. Indomethacin (CAS: 53-86-1) in 10-µM concentration was used as a positive control for COX-1 assays and Zileuton (CAS: 111406-87-2) in 3-µM concentration for 5-LO assays. 

Activities for assaying 11β hydroxysteroid dehydrogenase 1 (11β HSD1) and 11β hydroxysteroid dehydrogenase 2 (11β HSD2) were determined as previously described by Kratschmar et al. [75]. Briefly, lysates of HEK-293 cells stably expressing human 11β HSD1 were incubated with 200-nM cortisone (including 10-nM (1,2-^3^H)-cortisone), 500-µM NADPH, and the test substance. For 11β HSD2, lysates of HEK-239 cells stably expressing human 11β HSD2 were incubated with 50-nM cortisol (including 10-nM (1,2,6,7-^3^H)-cortisol), 500-µM NAD^+^, and the test compounds. Conversion of the radiolabeled substrate was determined and compared to enzyme activity in the control sample. 18β-Glycyrrhetinic acid (CAS: 471-53-4) was used as the positive control for both enzymes. Inhibitory activities for 3β hydroxysteroid dehydrogenase 1 (3β HSD1) and cytochrome P450 17A1 (CYP17A) were assayed as described before by Samadari et al. and Udhane et al. [76,77]. Activities were measured in cell-based assays using human adrenocortical NCI-H295R cells obtained from the American Type Culture Collection (ATCC; CRL-2128). The cells were treated with tritiated substrates and the product mix separated by thin-layer chromatography (TLC), and the resulting spots were subsequently densiometrically quantified. Trilostane (CAS: 13647-35-3) was used as the positive control. Inhibitory activities towards soluble epoxide hydrolase (sEH) were assayed using the purified enzyme as described by Wixtrom et al. and Morisseau et al. [78,79]. A baculovirus was used to transduce sEH into Sf9 insect cells, which were subsequently lysed and the enzyme purified using affinity chromatography. Enzyme inhibition could then be quantified using the purified sEH and substrate, which turned into a fluorophore by the latter, which can be read at 465 nm after excitation at 300 nm [80]. AUDA (CAS: 479413-70-2) was used as the positive control. Inhibitory activities towards 17β HSD4 were assayed according to the description in Schuster et al. [35]. Briefly, a plasmid coding for 17β hydroxysteroid dehydrogenase 4 (17β HSD4) was transformed into *E. coli* BL21 (DE3) Codon Plus RP (Stratagene). Subsequently, bacterial suspensions were prepared and incubated in the presence of 21-nM 17 β-estradiol (6,7-^3^H), 750-nM NAD^+^, and 10-µM test compound (1% DMSO final). After a defined incubation time, substrate and product were extracted using solid-phase extraction (SPE) and analyzed with RP-HPLC in a Beckman-Coulter system and online scintillation counting. Enzymatic conversion was calculated by integrating substrate and product peaks and calculating the percent inhibition relative to a control assay without an inhibitor (1% DMSO). Compound **19** (CAS: 1340482-23-6) from [72] served as the positive control.

### 4.6. Materials

Compounds **1**–**10** were purchased at TransMIT GmbH (PlantMetaChem, Gießen, Germany) with the following product numbers: **1**: P 036, **2**: H 031, **3**: D 017, **4**: A 020, **5**: D 018, **6**: S 025, **7**: P 037, **8**: T 017, **9**: P 064, and **10**: N 019. Purity was assessed by HPLC with a diode array detector (280 nm) found to be above 95% for all compounds (see Appendix A).

## Figures and Tables

**Figure 1 ijms-21-07102-f001:**
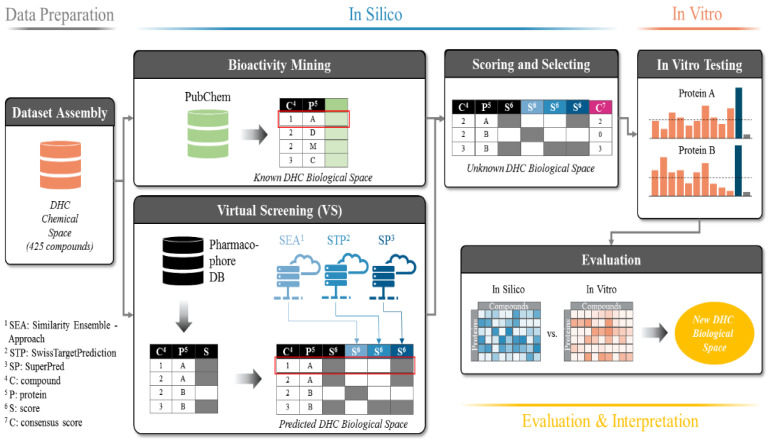
Workflow of the dihydrochalcone (DHC) target prediction campaign. The dataset is assembled (DHC chemical space) and used to retrieve corresponding bioactivity data from PubChem (known DHC biological space) and as input to inverse VS. First, the DHC chemical space is mapped onto the Pharmacophore DB (database) and the resulting matrix extended by the predictions of three individual target prediction servers: Similarity Ensemble Approach (SEA), SwissTargetPrediction (STP), and SuperPred (SP), resulting in the predicted DHC biological space. Activities already known from PubChem (the known DHC biological space) are then removed from the predicted DHC biological space, and the reduced matrix is scored according to the consensus predictions of the ligand-target interactions (unknown DHC biological space). Protein targets of the unknown DHC biological space are selected according to their consensus score (CS) and chosen for in vitro biological evaluation.

**Figure 2 ijms-21-07102-f002:**
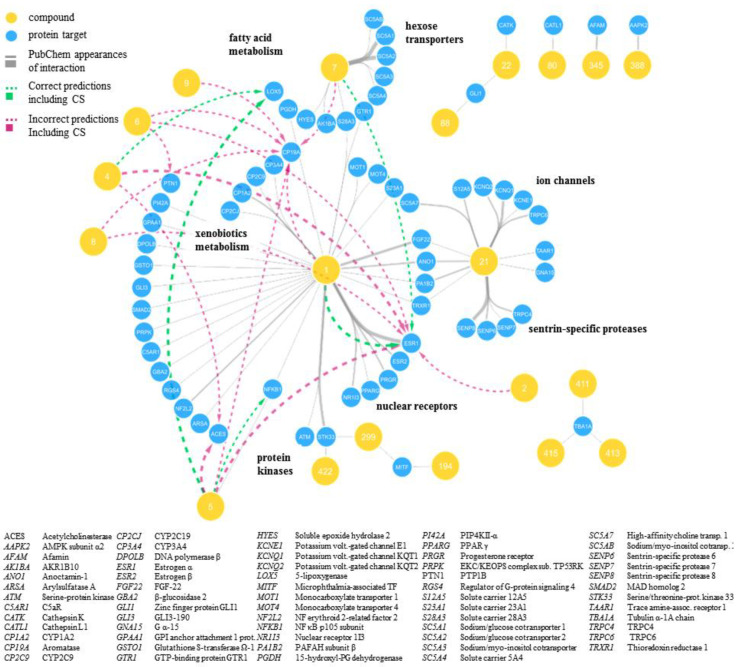
Known DHC biological space illustrated as a network. Shown are only DHCs with interactions reported in PubChem (grey edges), as well as the respective interactions predicted for these compounds (dashed green arrows for correct predictions and dashed magenta arrows for interactions that either proved incorrect in vitro or were not tested). Blue nodes indicate protein targets, and yellow nodes indicate compounds with respective compound numbers. Grey edges indicate known compound-target interactions, while the line thickness is proportional to the interaction weight (see Materials and Methods).

**Figure 3 ijms-21-07102-f003:**
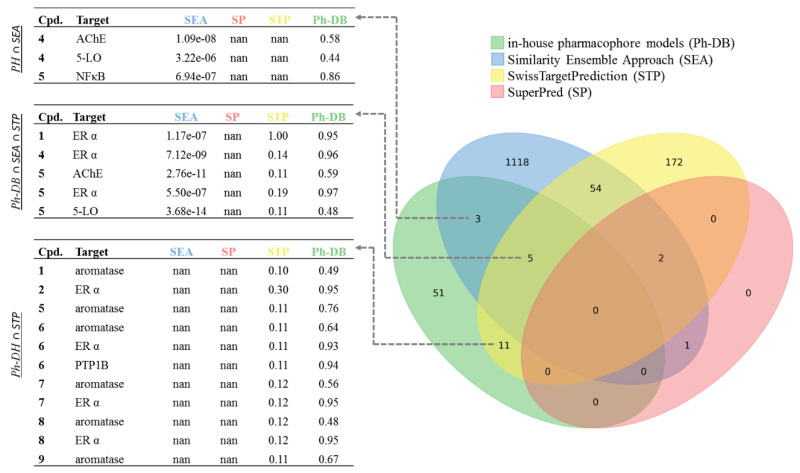
Predicted compound—target interactions (predicted DHC chemical space) illustrated as a Venn diagram. Predicted DHC chemical space was filtered for compounds **1**–**10** and a positive prediction by Ph-DB. Sets of predictions by any method are represented as ellipses (green: Ph-DB, blue: SEA, yellow: STP, and red: SP) and consents among different methods as overlaps/intersections, each with an integer indicating the size of the intersection. Values indicate the original model fit values; “not a number” (for short, “nan”) indicates no prediction by the respective tool. Fit values of SEA and SP are E-values similar as in the Basic Local Alignment Search Tool (BLAST), meaning the lower, the better the model fit [29,30,31]. Fit values for STP and Ph-DB are 0–1 normalized probability (STP) or relative pharmacophore fit scores (Ph-DB) [32,33].

**Figure 4 ijms-21-07102-f004:**
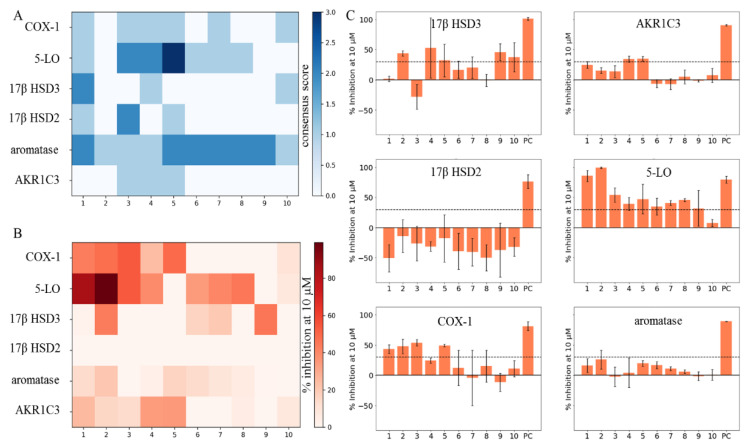
Comparison of an unknown DHC biological space (COX-1, 5-LO, 17β HSD3, 17β HSD2, aromatase, and AKR1C3) and actual in vitro test results of compounds **1**–**10**. (**A**) CSs of compounds **1**–**10** on all of the six targets of the unknown DHC biological space plotted as a heatmap. (**B**) Means (*n* = 3) of the percent inhibition at 10 µM (0–100%) of compounds **1**–**10** on all of the six targets of the unknown DHC biological space plotted as a heatmap. Observations with mean inhibition values smaller than 30% or relative standard deviations greater than 20% were regarded as inactive. (**C**) Bar charts of the six targets of the unknown DHC biological space showing compounds **1**–**10** with the respective means (*n* = 3) of the percent inhibition at 10 µM (0–100%) and the standard deviation. A cut-off of 30% inhibition at 10 µM was chosen (black dashed line), for separating the active from inactive observations. Dimethylsulfoxide (DMSO) was used to measure the baseline enzyme activities on which the samples were normalized (not shown), and positive controls (PC) were used as indicated in the Materials and Methods and in Appendix A.

**Table 1 ijms-21-07102-t001:** Compounds **1**–**10**, which were used for the biological evaluation.

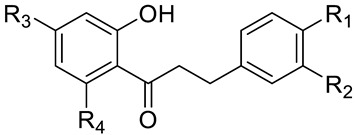					
**No.**	**Name**	**R_1_**	**R_2_**	**R_3_**	**R_4_**
1	phloretin	OH	H	OH	OH
2	3-OH-phloretin	OH	OH	OH	OH
3	2′,6′-dihydroxy-4′-methoxy DHC	H	H	OMe	OH
4	asebogenin	OH	H	OMe	OH
5	calomelanen	OMe	H	OMe	OH
6	sieboldin	OH	OH	O-Glc ^1^	OH
7	phloridzin	OH	H	OH	O-Glc ^1^
8	trilobatin	OH	H	O-Glc ^1^	OH
9	phloretin-2′-xyloglucoside	OH	H	OH	O-Rut ^2^
10	neohesperidin DHC	OMe	OH	O-Neo ^3^	OH

^1^ Glc: glucose (O-β-d-glucosyl). ^2^ Rut: rutinose (6-O-(α-l-rhamnosyl)-d-glucos-1-O-β-yl). ^3^ Neo: neohesperidose (2-O-(α-l-rhamnosyl)-d-glucos-1-O-β-yl). DHC: dihydrochalcone.

**Table 2 ijms-21-07102-t002:** Twelve frequently predicted targets for DHC chemical space assessed according to selection criteria I–IV and a final selection statement.

Candidate Target	Selection Criterion I ^1^	Selection Criterion II ^2^	Selection Criterion III ^3^	Selection Criterion IV ^4^	Selected
17β HSD2	n.a. ^5^	5th (CS = 3) 12th (CS = 2)	1-ER α/β	Yes	Yes
17β HSD3	n.a. ^5^	4th (CS = 3) 7th (CS = 2)	1-ER α/β	Yes	Yes
5-LO	4 (CS = 2) 5 (CS = 3)	3rd (CS = 3) 3rd (CS = 3)	1-PGDH	Yes	Yes
AChE	4 (CS = 2) 5 (CS = 3)	1st (CS = 3) 10th (CS = 2)	n.a. ^5^	No	No
AKR1C3	n.a. ^5^	6th (CS = 2)	1–AKR1B10	Yes	Yes
Aromatase	1 (CS = 2) 5 (CS = 2) 6 (CS = 2) 7 (CS = 2) 8 (CS = 2) 9 (CS = 2)	1st (CS = 2)	1-aromatase 1-ER α/β 1-several CYPs	Yes	Yes
COX-1	n.a. ^5^	4th (CS = 3)	1-PGDH	Yes	Yes
ERα	1 (CS = 3) 2 (CS = 2) 4 (CS = 3) 5 (CS = 3) 6 (CS = 2) 7 (CS = 2) 8 (CS = 2)	2nd (CS = 3) 2nd (CS = 2)	1-ER α/β	Yes	No
ERβ	n.a. ^5^	5th (CS = 3) 6th (CS = 2)	1-ER α/β	Yes	No
NF-κB	n.a. ^5^	8th (CS = 2)	1-NF-κB 5-NF-κB	No	No
PPARγ	n.a. ^5^	9th (CS = 2)	1-PPARγ	No	No
PTP1B	6 (CS = 2)	10th (CS = 2)	n.a.	Yes	No

^1^ Targets that were predicted with high consensus scores (CSs) for compounds **1**–**10**. ^2^ Most frequently predicted targets with high CSs for the DHC chemical space. ^3^ Prediction consistency: similar or associated targets that are being predicted or similar bioactivities that were already reported. ^4^ Availability of a suitable assay. ^5^ Not applicable. NF-κB: nuclear factor κB.

**Table 3 ijms-21-07102-t003:** In vitro inhibitory activities of compounds **1**–**10** towards targets of the DHC biological space, expressed as percent inhibition (0–100%) at 10-µM compound concentration relative to the DMSO mock control. Shown is the mean of three independent experiments (*n* = 3) plus/minus the standard deviation. Different compounds were used as positive controls (PC), as indicated in the Materials and Methods. Highly negative values of 17β HSD2 assays are believed to be technical artefacts, as enzyme activation seems unlikely.

Compound	Aromatase	17β HSD2	17β HSD3	AKR1C3	5-LO	COX-1
**1**	13.8 ± 2.0	−50.7 ± 22.7	1.7 ± 4.5	24.8 ± 5.9	85.4 ± 9.3	43.5 ± 7.2
**2**	21.1 ± 11.7	−14.1 ± 27.1	43.8 ± 4.7	15.5 ± 4.9	99.2 ± 1.2	48.1 ± 12.0
**3**	−1.0 ± 12.0	−26.3 ± 28.8	−28.0 ± 20.5	13.8 ± 10.0	54.1 ± 11.6	53.9 ± 5.3
**4**	3.5 ± 19.0	−31.2 ± 8.1	52.7 ± 49.6	34.4 ± 5.1	39.2 ± 11.1	24.4 ± 4.2
**5**	17.0 ± 2.0	−17.6 ± 39.0	32.1 ± 27.2	35.2 ± 3.8	47.2 ± 24.2	49.5 ± 1.9
**6**	13.8 ± 4.4	−39 ± 29.9	16.7 ± 14.5	−6.1 ± 6.8	34.8 ± 14.2	12.34 ± 29.0
**7**	9.4 ± 3.5	−40.5 ± 22.9	20.2 ± 17.7	−7.2 ± 8.9	40.8 ± 4.1	−4.2 ± 45.6
**8**	5.9 ± 3.5	−49.8 ± 21.4	−0.6 ± 10.0	5.3 ± 11.5	45.5 ± 2.7	15.3 ± 26.4
**9**	0.67 ± 4.0	−37.1 ± 44.3	45.8 ± 14.2	−1.3 ± 1.9	31.8 ± 29.4	−11.4 ± 15.1
**10**	0 ± 5.8	−32.2 ± 15.4	37.5 ± 23.9	7.4 ± 11.8	7.7 ± 6.0	11.1 ± 13.2
**PC**	70.2 ± 0.5 *	76.1 ± 11.4 ^†^	101.2 ± 2.4 ^‡^	90.5 ± 1.2 ^§^	79.26 ± 5.95 ^¶^	81.3 ± 7.5 ^#^

* 10-nM anastrozole (CAS: 120511-73-1). ^†^ 1-µM ML376 (CAS: 1340482-23-6). ^‡^ 1-µM compound 24 (CAS: 873206-61-2) [34]. ^§^ 1-µM compound 2–9 (CAS: 745028-76-6) [35]. ^¶^ 3-µM zileuton (CAS: 111406-87-2). ^#^ 10-µM indomethacin (CAS: 53-86-1) (see Appendix A).

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
