# Peer review of "Finding New Molecular Targets of Familiar Natural Products Using In Silico Target Prediction"

_ijms, 2020, doi:10.3390/ijms21197102_

Round 1
Reviewer 1 Report
This work deals with the investigation of New Molecular Targets of known Natural or semisynthetic Products Using In Silico technologies.
Its an interesting and well written manuscript, that, in my opinion could be published in IJMS. The idea of predicting new activities of natural compounds is interesting, and probably could help on the design - semi-synthesis of new ones with better activity and selectivity.
Just a few questions for the authors:
- Can this methodology be expanded for the design of new compounds with better activity?
- Usually, natural compounds are very complex, or at least more complex than DHCs. In addition, in the literature there many examples of known DHCs in order to “educate” the workflow. Could this “technology” be used in more complex and not so common natural compounds?
Author Response
We thank reviewer 1 for his / her time and helpful comments on our manuscript.
Regarding the two questions to the authors:
- Virtual target prediction could also be used on newly designed compounds and some of the applied methods, such as pharmacophore modelling could also be used to optimize activity. However, this is not the main focus of our workflow.
- Yes, this method could also be used on more complex structures than DHCs. The limits of applicability of the workflow rest on the similarity of the query molecules to known actives. For example a molecule much larger than all known actives would likely not be identified as a virtual hit, by the model
Regarding the second question, we added some text to the discussion section of the manuscript:
"
Fourth, the established in silico prediction workflow can be applied to several other compound classes alike DHCs, even much more complex structures. The limit of the applicability is the similarity of the query compounds to the reference compounds used by the prediction platforms (known actives from the literature). Compounds that significantly differ from the currently known biologically active chemical space, e.g. compounds with a very high molecular weight, may not be identified as virtual hits by the workflow.
No further changes were made.
Reviewer 2 Report
The authors improved or corrected the manuscript in all requested parts with the exception of the addition of the IC50 and the relative curves of the most interesting compounds. They justified this decision due to the COVID-19 situation and the extension of time. Although this addition would increase the overall quality of the manuscript, however this manuscript can be published as article on the International Journal of Molecular Sciences in present form
Author Response
We thank reviewer 2 for her / his time and useful comments. We agree that the IC50 and the relative curves of the most interesting compounds would have added value to the study. We hope that the current situation will improve within the next months and that all of us will be able to work "normally" again in the labs.
This manuscript is a resubmission of an earlier submission. The following is a list of the peer review reports and author responses from that submission.
Round 1
Reviewer 1 Report
The authors describe a drug target identification workflow by performing a virtual screen of 425 dihydrochalcone and similar compounds against targets predicted by four computational prediction tools and taking the consensus scores for each. Three of the tools are publicly available web resources, while the fourth is a pharmacophore-based virtual screen developed by the authors. They then followed this by validating ten of these dihydrochalcone compounds against six targets in 10 micromolar inhibition assays. It appears from Figure 2 that five interactions predicted by the workflow were verified by assay and thirteen incorrect predictions resulted from the biological screen(?).
The language in the text is very colloquial, e.g., line 66: "nowadays" and line 88: "On the other hand", and needs some copy editing throughout("that that" on line 67, the Bibliography has two sets of numbers, etc.). Line 81: "consent" should be "consensus". Lines 82-84: It is unclear what you are saying.
Known protein targets are excluded from the results, but what is the rationale? Figure 1 gives a better description than the text.
Reviewer 2 Report
The authors of the present article describe the results of an in silico target prediction. In particular, they developed an interesting in silico workflow consisting of four independent target prediction tools by using open-source platforms and they applied this computational approach on the natural compounds dihydrochalcones. This approach allowed them to identify i) four previously unreported protein targets for this class of natural compounds such as 5-lipoxygenase, cyclooxygenase-1, 17β-hydroxysteroid dehydrogenase 3, and aldo-keto reductase 1C3 and ii) new leads to use as starting point for the development of new inhibitors of the new identified protein targets. The obtained results highlight that this strategy could be useful for the prediction of new molecular targets and for the identification of new leads also for other classes of known natural and synthetic compounds.
The manuscript is detailed and well written, but this paper is to be published as article on the International Journal of Molecular Sciences after one major revision and some minor revisions:
Major revision:
The sentences in the section discussion at page 10 "Second, to remain with polypharmacology, e.g.4 and 5 could serve as lead to develop dual inhibitors of AKR1C3 and 5-LO. Analogously, 2 could be used as a starting point for the development of a dual 17β HSD3/5-LO inhibitor." are not completely correct. The compound 5 is more active against COX1 than AKR1C3. On the other hand, the best activity of compound 4 is against 17β HSD3. The compound 2 is more active against 5-LO and COX1 than 17β HSD3.
However, the authors should report the IC50 and the relative curves of the most interesting compounds. On the basis of the IC50 values, the authors can define the polypharmacology of these dihydrochalcones and some compounds can be cited as new possible leads for the development of new inhibitors of the new identified protein targets. Therefore, the authors have to rewrite a part of the paragraph "discussion" considering the IC50 values of the most interesting compounds.
Minor revisions:
- At page 2, line 50, the keyword “polyphamracology” should be corrected with “polypharmacology”
- At page 5, line 152, the sentence “(summarized for twelve frequently predicted targets in Table 2): First, interactions of 1 – 10 that were….” should be corrected with “(summarized for twelve frequently predicted targets in Table 2). First, interactions of 1 – 10 that were….”
- The authors should define in Table 2 the acronym “n.a.”
- At page 10, line 289, the sentence “The results can now be utilized in manifold ways: First, our findings….” should be corrected with “The results can now be utilized in manifold ways. First, our findings….”
- In the section “References” the authors should remove the double numbering present in all references.